# Stationary Traffic as a Factor of Tourist Destination Quality and Sustainability

**Robert Maršanic** [1], **Edna Mrnjavac** [2], **Drago Pupavac** [3],* and **Ljudevit Krpan** [4]

1 Department of Sustainable Mobility and Logistic, University North & Road Administration Primorje and Gorski Kotar County, 51000 Rijeka, Croatia; rmarsanic@unin.hr
2 Department of Tourism, Faculty of Tourism and Hospitality Management, 51410 Ika, Croatia; ednam@fthm.hr
3 Transport Department, Polytechnic of Rijeka, 51000 Rijeka, Croatia
4 Department of Sustainable Mobility and Logistic, University North & Primorje and Gorski Kotar County, 51000 Rijeka, Croatia; ljkrpan@unin.hr
* Correspondence: drago.pupavac@veleri.hr; Tel.: +385-989-071-218

**Abstract:** Since the Republic of Croatia is one of the most popular European and world tourist destinations, the aim of this paper is, from the user's ($n$ = 596) point of view, to research the importance of stationary traffic in tourist destinations. The purpose of this paper is to point out the possibilities of improving the tourist destination quality and sustainability through an adequate parking service. In order to corroborate constructed scientific hypotheses, a larger number of scientific methods were used from which a polling method, analysis and synthesis method, descriptive statistics method, $t$-test, and analysis of variance (ANOVA) should be singled out. The major finding of this paper indicates a relatively big importance of stationary traffic (M = 6.51; SD = 2.21) as an element of tourist destination quality. Moreover, regarding the quality of tourist destination, the results of this paper suggest that the parking space availability is more important than the way parking or parking payment are organized. Between the experienced parking problem in a tourist destination and age on one side and evaluation of the importance of stationary traffic as an element of tourist destination quality on the other side, a statistically important connection was established. Gained knowledge can be particularly helpful to hotel industry managers but also to traffic managers whose duty is to provide an adequate number of parking spaces in tourist destinations.

**Keywords:** stationary traffic; parking; hotel industry; tourist destination; quality; sustainability

## 1. Introduction

The Republic of Croatia is a tourist country and one of the most popular European tourist destinations. In 2019, there were 19.6 million tourist arrivals, including 2.2 million of domestic tourists and 17.4 million of foreign tourists. Altogether, 91.2 million tourist nights were recorded. Competition of tourist destinations on the European and world market is increasing every year [1]. Most tourist destinations use quality and sustainability development as one of the important strategies to increase their competitiveness in international tourist markets [2]. Destination quality increases tourist satisfaction and influences tourists' destination loyalty [3–5] and may affect tourists' decision to revisit the destination [6]. Destination loyalty signifies more recommendations to other people, more visitors, more employees, larger business growth, and more revenue for the hotel industry [7]. Destination loyalty also means more cars since foreign tourists tend to arrive in Croatia by car [8]. A crowded parking lot [9] and the lack of parking spaces can erode tourist satisfaction and tourists' destination quality and sustainability. It is important for destination managers to concentrate on the quality and sustainability of tourist destinations. Efficient organization and management of tourist supply in tourist destinations should inevitably also include stationary traffic as an important factor of tourist destination attractiveness and sustainability.

Parking in tourist destinations has become a first-class problem since traffic, which actually enabled the development of many tourist destinations, has been a growing limiting factor of tourist destination quality and sustainability. It is real to expect that problems of stationary traffic in tourist destinations will further increase, therefore an efficient solution to this problem requires new ideas and an interdisciplinary approach [10]. Under conditions when even 64% of total tourists come in a passenger car, and this is the case in the Republic of Croatia [11], it becomes utterly important to provide sufficient areas for a standstill at locations of visitors' particular importance for both every single destination and holders of tourists offering single elements. Therefore, the parking belongs to the services segment that make up a hotel product on a micro level representing an interruption in the process of tourist movement for the reason of staying at a tourist destination. Tourist destinations, when exposed to the impact of numerous and different factors, in providing parking spaces [12] for their guests are guided by different criteria and methods to solve this problem.

Accordingly, in this paper, several scientific hypotheses are represented. Research for the purpose of this paper was conducted in typical tourist auto-destinations such as the County of Primorje and Gorski kotar and Istria County that are also one of the most developed tourist and economic regions of the Republic of Croatia. These two counties in 2019 realized 7.46 million arrivals and 41.71 million nights, which make 38.1% in total arrivals and 45.7% in total tourist nights [13]. Research covered guests from 17 hotels and specifically: Four hotels in the area of the city of Rijeka, four hotels in the area of the town of Opatija, one hotel in the town of Krk, one hotel in the municipality of Omisalj (the island of Krk), six hotels in the town of Crikvenica, and one hotel in the town Rabac (Istria).

## 2. Theoretical Background

### 2.1. Literature Review

A tourist destination is a reason to travel [14], and the quality factor that makes up the tourist destination product is the reason for both domestic and foreign tourists to come and visit. The competitive strength of a tourist destination depends on the quality of factors it consists of. The task of quality factors of a tourist destination is double (cf. Figure 1).

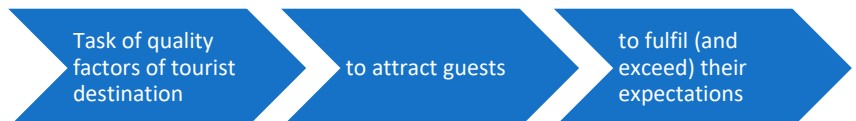

**Figure 1.** The task of quality factors of a tourist destination [15].

In the study of tourist destination factors Cetinski and Sugar [16] single out:

- Attractions (nature-beaches, climate, sunny days, sea, low-priced accommodation, historical monuments, stories and legends, famous persons, cultural events, etc. [17];
- accommodation and catering capacities (hotels, restaurants, apartments, resorts, camps, private houses, family run farms in the neighbourhood) [18,19];
- programs (events, concerts, exhibitions, workshops–local culture, trips, sport and recreation, wellness, entertainment–night life, services, attractions, and programs for children, etc.) [14];
- shopping possibilities (shopping) [20];
- peace and safety [21];
- environmental awareness (cleanness of the sea, beaches, air, healthy food, less noise) [22];
- communications (traffic and connections) [23];
- information (timeliness, entirety, availability) [24];
- orderliness and cleanness of the city [25];
- value for money (value for money) [26] and for effort (value for effort) [27];
- hospitality and knowledge of foreign languages [28].

It is evident that the authors do not point out stationary traffic as an important factor for tourist destination quality or that they marginalize it if they possibly classify it indirectly as the communication factor (traffic and connections). By failing to point out stationary traffic, its role as a tourist destination factor is being marginalized and accordingly no sufficient attention is being given to it. In his research of tourist destination quality, Holjevac [29] entirely neglects stationary traffic as a tourist destination quality factor. A similar situation is found in other countries of Southeast Europe. For instance, in the Republic of North Macedonia, [30] with regards to traffic as a tourist destination element, only the availability of a destination to visitors is highlighted, i.e., traffic infrastructure and possibility of local transport at the destination. Similar examples can also be found in other countries of the region [31]. Interest for stationary traffic as an important element for tourist destination quality in the Republic of Croatia is rather recent [32]. According to the study by Rudancic-Lugaric [33], stationary traffic as an element of tourist destination quality got the lowest grades in renowned Croatian tourist destinations such as Opatija (2.67) and Porec (2.78). An evaluation was made on the Likert scale from 1 to 7. Stationary traffic and all other quality factors of a tourist destination should be based on sustainable development [34]. According to Crane and Matten [35], sustainability refers to the long-term maintenance of system according to environmental, economic, and social considerations. Resources within tourism are limited and the main goal of the Europe tourist destinations must be high quality and sustainable tourism [36]. Sustainability is *condition sine qua non* of the high quality of a tourist destination.

Contemporary tourist trends are characterized by a major role of cars during travelling. An increasing number of tourists choose to take advantage of weekends and long weekend holidays to take small trips [37]. These shorter stays involve more visits, more cars, and more parking problems. Therefore, it is comprehensible that one of the elements of tourist destination quality is parking services, that is, providing space for stationary cars of both domestic and foreign tourists since using a car means movement and standstill rotation. Destinations vary with regards to the tourist offer, but also as to whether they recognize the parking problem. The majority solve this problem by developing public parking spaces for money. Such an approach is understandable since seasonality is an important feature of Croatian tourism. Only a small number of destinations resorts to developing car parks and parking facilities and these are usually towns with a larger number of local inhabitants that generate a certain level of a whole-year parking demand.

Nevertheless, stationary traffic has rarely been the subject of scientific research and when it is, a traditional parking approach prevails [38] that does not at all or only marginally takes into account contemporary development concepts in the surroundings. Tourism, and generally the development policy, are usually focused on increasing basic accommodation capacities. Tourists' needs for stationary cars at the destination are one of the basic determinants of the traffic and tourist demand. Studies on the satisfaction of destination's visitors show a lower satisfaction level if there are no sufficient parking spaces [39]. It may be concluded that the insufficient capacity and incorrect location of parking spaces increases tourists' dissatisfaction and also the quality evaluation of a tourist destination. To meet the traffic-tourist demand for movement and standstill in quantitative and qualitative terms is a significant determinant of a wider context of mobility at a tourist destination. Therefore, it is particularly important to replace the traditional parking approach, which is based on a separate research of this occurrence, with a holistic approach. The parking problem particularly afflicts hotel companies in tourist car-destinations to which tourists come with road vehicles, especially by passenger cars, even though at the same time the parking problem in the "aeroplane" tourist destinations should not be neglected due to the developed rent-a-car practice.

Trends in the structures of traffic modalities in tourist flows do not indicate a significant decrease of the role of road traffic [40]. The share of road traffic in tourist flows on the world level is in regression in favour of air traffic [41], but passenger cars will be very hard to substitute in the near future since there is no transport means with similar

features and moreover, it is not very likely for it to appear. Changes will go in directions of environmental forms of drive, but all measures aimed at the decreased use of cars have produced no results. Therefore, parking spaces in hotel companies are expected to last further on or even to increase so that new concepts based on information and communication technologies shall be required to solve them efficiently. Some authors [42] see the parking services as a dimension of added value concept in hotel products, which has a positive and significant influence on guests' overall satisfaction and their return to the same hotel. Moreover, quite often hotel companies do not have their own parking capacities, but they lease a certain, most commonly insufficient number of parking spaces at the nearest city parking areas. Furthermore, more than often guests themselves need to look for a place to park at the tourist destination and park at the places that are not foreseen for these purposes (irregular, that is, illegal parking), which leads to numerous other problems and possible inconveniences.

*2.2. Concept of Research*

Hotel companies operating the whole year have a tremendous advantage in providing parking spaces for their guests over hotels with distinct business seasonality. Regularity and a stable level of income justify bigger investments in quality parking capacity, most often in the own car park, if space and urban requirements at the destination allow them to do so. Adequate "caring for", i.e., providing for parking cars of their own guests by hotels decreases the pressure on public parking areas and opens up the possibility for guests to move around the destination more easily whether with a passenger car or without it. Then, they can give way to their planned activities (recreation, sports, entertainment, resting) and programs they intended to visit (historical and cultural locations, culinary and wine offer) and not put their plans into the context of the problem where will they park, will they find a free parking space, what will they do if this does not happen, how long can they leave their car in the parking space, how much will they pay for the parking space?

Parking has become a major problem [43] in many tourist destinations [44] and demands an integrated approach to transportation and tourism planning [45]. For example, stationary traffic is one of the main problems of the famous cultural tourist destinations in Spain, Toledo. They try to solve this problem through the progressive construction of paid underground parking areas, free outdoor parking areas far from the historic center, traffic restriction in some areas and in some cases, with a system of automatic bollards [46]. Providing parking space is one of the most important elements in the complete offer by hotels and tourist destinations, although not a decisive one [47]. In hotel companies that were subject and relevant to this research a parking service was provided in one of the following four ways [48]: (1) On-street parking, (2) off-street parking, (3) garage facilities, and (4) in some other way. Usta, Berezina, and Cobanoglu [49] examined the hotel attributes as perceived by travelers and found that free parking is on the top third place among fifty attributes. According to a recent research, 55% of US travelers consider free parking very important and 33% somewhat important when choosing a hotel for a business trip [50].

In line with the previously mentioned, a conceptual research model is set (cf. Figure 2).

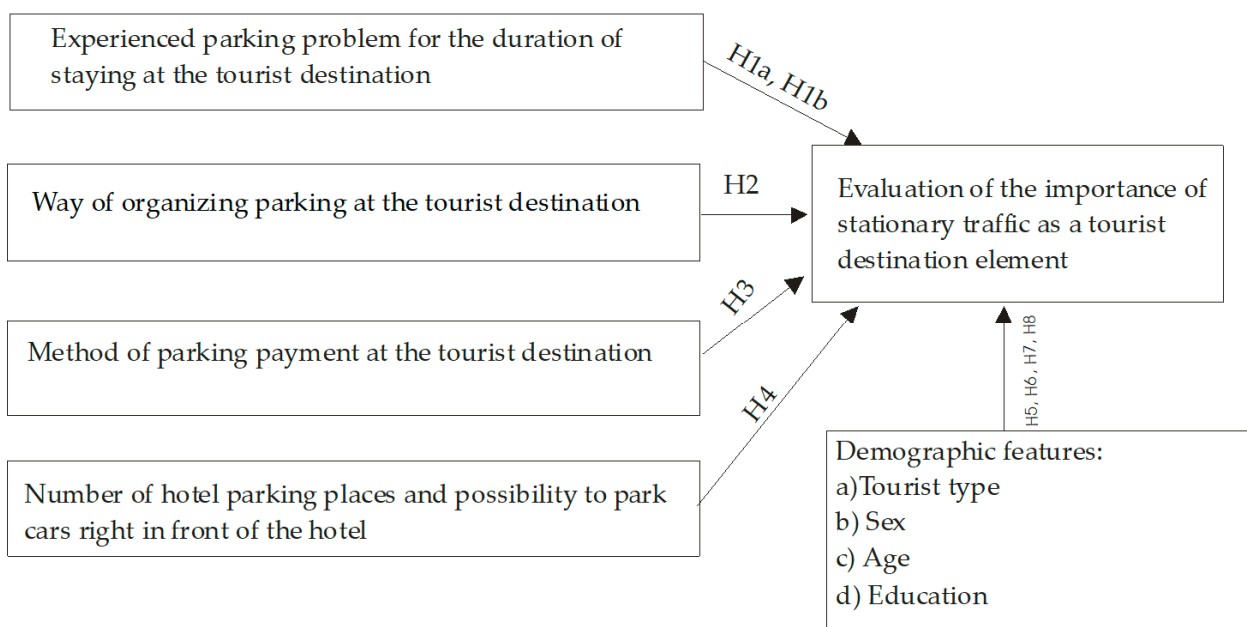

**Figure 2.** Conceptual research model of the importance of stationary traffic as the quality factor of a tourist destination.

Based on the conceptual model, it can be concluded that an evaluation of the importance of stationary traffic as a factor of tourist destination quality depends on: (1) Experienced parking problem during the stay at a tourist destination, (2) way of organizing parking at the tourist destination, (3) method of parking payment at a tourist destination, and (4) number of parking spaces and a possibility to park a passenger car right in front of the hotel at a tourist destination.

Accordingly, the following hypotheses were constructed:

**Hypothesis 1a (H1a).** *Tourists that had not experienced a parking problem gave a higher grade to the importance of stationary traffic as a factor of tourist destination quality.*

**Hypothesis 1b (H1b).** *Tourists whose parking problem had been solved in a satisfactory way gave a higher grade to the importance of stationary traffic as a factor of tourist destination quality than tourists whose problem had not been solved in a satisfactory way.*

**Hypothesis 2 (H2).** *The way of organizing parking impacts the grade of the importance of stationary traffic as a factor of tourist destination quality.*

**Hypothesis 3 (H3).** *The way of parking payment impacts the grade of the importance of stationary traffic as a factor of tourist destination quality.*

**Hypothesis 4 (H4).** *The number of parking spaces and a possibility to park the car right in front of the hotel impact the grade of the importance of stationary traffic as a factor of tourist destination quality.*

**Hypothesis 5 (H5).** *Foreign tourists give a higher average grade to the importance of stationary traffic as a factor of tourist destination quality.*

**Hypothesis 6 (H6).** *Between men and women there are no statistically significant differences in evaluating the importance of stationary traffic as a factor of tourist destination quality.*

**Hypothesis 7 (H7).** *Between guests of different age there are no statistically significant differences in evaluating the importance of stationary traffic as a factor of tourist destination quality.*

**Hypothesis 8 (H8).** *Between guests of different education degrees there are no statistically significant differences in evaluating the importance of stationary traffic as a factor of tourist destination quality.*

## 3. Materials and Methods

Field research was based on a survey questionnaire filled by hotel guests. A questionnaire with 12 questions was used as the research instrument. The questionnaire was divided into two parts. The first part consisted of four questions on socio-demographic characteristics of the respondents (sex, age, education, and the state of origin). The second part of the questionnaire was comprised of eight questions focusing on the availability and quality of parking in tourism destinations and hotels. Information were gathered on: Existence of a parking problem at the hotel, sufficiency of parking spaces at the hotel, ways of parking, successfulness of managers in solving parking problems, if any, modalities of parking payment and the importance of parking as a factor of the tourist destination quality. A survey was conducted in the period from 15 July until 1 September 2019. The survey response rate was 21%, and 713 questionnaires were collected out of which 596 questionnaires were correct. The survey method was chosen because filling in the survey questionnaire with answers offered in advance requires a minimum guests' involvement, thus it was considered the optimum way to gain as big a number of filled-in forms as possible, which was one of the major goals. The survey was anonymous, guests had no obligations whatsoever in filling in the survey. Survey questionnaires were offered in three languages: Croatian, English, and German. The study covered guests from 17 hotels, specifically: Four hotels in the area of the city of Rijeka, four hotels in the area of the town of Opatija, one hotel in the town of Krk, one hotel in the town of Omišalj (island of Krk), six hotels in the town of Crikvenica, and one hotel in Rabac (Istria). The statistical evaluation of collected data has been done with the support of program *Statistica*.

The sample (*n* = 596) was made up of 357 (59.9%) men and 239 (40.1%) women. One hundred and forty domestic guests made up 23.49% of survey participants, and 456 foreign guests made up 76.51% of survey participants. Austrians (16.61%), Germans (10.9%), Italians (10.4%), and Slovenes (9.89%) were in the forefront of foreign guests. The largest number of survey participants was between 36 and 49 years old, that is, 182 of them or 30.5 %, followed by survey participants at the age from 50 to 64, 176 of them or 29.53% (cf. Table 1).

**Table 1.** The structure of survey participants.

| Characteristics | Domestic Tourist | | Foreign Tourist | | Total | |
|---|---|---|---|---|---|---|
| | *n* | % | *n* | % | *n* | % |
| Sex | | | | | | |
| Male | 85 | 60.71 | 272 | 59.65 | 357 | 59.9 |
| Female | 55 | 29.39 | 184 | 40.35 | 239 | 40.1 |
| Total | 140 | 23.49 | 456 | 76.51 | 596 | 100.00 |
| Age | | | | | | |
| 18–25 | 21 | 15.00 | 49 | 10.75 | 70 | 11.74 |
| 26–35 | 33 | 23.57 | 99 | 21.71 | 132 | 22.15 |
| 36–49 | 45 | 32.14 | 137 | 30.04 | 182 | 30.54 |
| 50–64 | 34 | 24.29 | 142 | 31.14 | 176 | 29.53 |
| 65+ | 7 | 5.00 | 29 | 6.36 | 36 | 6.04 |
| Total | 140 | 23.49 | 456 | 76.51 | 596 | 100.00 |

**Table 1.** *Cont.*

| Characteristics | Domestic Tourist | | Foreign Tourist | | Total | |
|---|---|---|---|---|---|---|
| | *n* | *%* | *n* | *%* | *n* | *%* |
| Level of Education | | | | | | |
| Elementary school | 5 | 3.57 | 28 | 6.14 | 33 | 5.54 |
| Secondary school | 56 | 40.00 | 160 | 35.09 | 216 | 36.24 |
| Faculty | 70 | 50.00 | 238 | 52.19 | 308 | 51.68 |
| PhD | 9 | 6.43 | 30 | 6.58 | 39 | 6.54 |
| Total | 140 | 23.49 | 456 | 76.51 | 596 | 100.00 |

Source: Authors, based on results obtained by surveying guests.

## 4. Results and Discussion

Survey participants were supposed to evaluate with grade from 1 to 10 the importance of stationary traffic as a factor of tourist destination quality. Based on the gathered data (cf. Table 2), a brief descriptive analysis of their evaluation was made.

**Table 2.** Descriptive statistics of evaluation of the importance of stationary traffic as a factor of tourist destination quality.

| | Grade |
|---|---|
| MEAN case 1-596 | 6.51 |
| MEDIAN case 1-596 | 7 |
| SD case 1-596 | 2.21 |
| VALID_N case 1-596 | 596 |
| SUM case 1-596 | 3883 |
| MIN case 1-596 | 1 |
| MAX case 1-596 | 10 |
| _25th% case 1-596 | 5 |
| _75th% case 1-596 | 8 |

Source: Authors, based on results obtained by surveying guests.

The average grade of stationary traffic as a factor of tourist destination quality is (M = 6.51; SD = 2.21). A high median value (7) that divides the set into two equal parts shows great importance of stationary traffic as a factor of tourist destination quality. This assertion is supported also by the fact that the first quartile is made up of survey participants who gave the importance of stationary traffic as a factor of tourist destination quality a grade less than 5, and the last quartile is made up of survey participants who gave the importance of stationary traffic as a factor of tourist destination quality a grade higher than 8. The remaining 50% of survey participants gave the stationary traffic as a factor of tourist destination quality a grade ranging from 5 to 8.

Hereinafter, the constructed hypotheses are tested.

On the grounds of the conducted research it was established that every fourth guest had some kind of a problem with the parking service at a tourist destination (cf. Table 3).

**Table 3.** Parking problem at a tourist destination.

|  | Count | Cumulative-Count | Percent | Cumulative-Percent |
|---|---|---|---|---|
| Yes | 154 | 154 | 25.83893 | 25.8389 |
| No | 442 | 596 | 74.16107 | 100.0000 |
| Missing | 0 | 596 | 0.00000 | 100.0000 |

Source: Authors, based on results obtained by surveying guests.

On the grounds of the conducted *t*-test, the constructed hypothesis can be accepted with 95% reliability suggesting that guests, who did not have a parking problem, value with a higher grade the importance of stationary traffic as a factor of the tourist destination quality. Research results indicate that survey participants that did not experience a parking problem at a tourist destination attach greater importance to stationary traffic as a factor of tourist destination quality (M = 6.87; SD = 2.08) than tourists that experienced a parking problem at a tourist destination (M = 5.49; SD = 2.26; *t* = 6.9; *p* = 0.00).

Management of the total quality of tourist destination quality dictates that the best service is to be provided at the first attempt [51,52]. However, as errors are possible, then it is necessary to act promptly and remove the cause of user's dissatisfaction since, otherwise service quality will be permanently undermined. As for solving the parking service problem, out of 154 users, i.e., 105 users or 68.18% said that they were satisfied with the way hotel management solved their problem, and 49 users or 31.82% said they were not satisfied with the way hotel management solved their problem. Testing the H1a hypothesis was also carried out with a *t*-test. Although research results indicate that survey participants whose parking problem was solved in a satisfactory way (M = 5.6; SD = 2.2) give slightly higher grades to the importance of stationary traffic as an element of the tourist destination than those survey participants whose parking problem was not solved in a satisfactory way (M = 5.24; SD = 2.39; *t* = 0.93; *p* = 0.35), the constructed hypothesis failed to reject.

In order to evaluate the existence of differences between the ways of organizing parking [53] and evaluation of the importance of parking as an element of tourist destination quality, the ANOVA method was applied (cf. Table 4).

**Table 4.** Difference between organizing a parking service and evaluation of the importance of parking as an element of tourist destination quality with the applied ANOVA method.

|  | Organize_P | Grade-Mean | Grade-Std. Err. | Grade −95.00% | Grade +95.00% | *n* |
|---|---|---|---|---|---|---|
| 1 | on-street | 6.491166 | 0.131611 | 6.232685 | 6.749647 | 283 |
| 2 | off-street | 6.279661 | 0.203818 | 5.879366 | 6.679956 | 118 |
| 3 | garage | 6.711538 | 0.217104 | 6.285151 | 7.137926 | 104 |
| 4 | other | 6.670330 | 0.232094 | 6.214502 | 7.126157 | 91 |

Organize_P; LS Means Current effect: F(3, 592) = 0.87781, *p* = 0.45229 Effective hypothesis decomposition.

Even though guests that used parking services in the hotel garage facilities gave the highest average grade (M = 6.71) to the importance of stationary traffic as an element of tourist destination quality, the constructed hypothesis is rejected. Specifically, differences in arithmetic means between guests that used a different parking service are not statistically significant.

In hotel companies that were subject and relevant to this research a parking service was: (1) Free, (2) included in a hotel service price or (3) subject to a separate payment. In order to research the existence of differences in the way of parking service payment and evaluation of the importance of parking as an element of tourist destination quality, the ANOVA method was applied (cf. Table 5).

**Table 5.** Difference in the way of parking service payment and evaluation of the importance of parking as an element of tourist destination quality by applying the analysis of variance (ANOVA) method.

| | P_Paid | Grade-Mean | Grade-Std. Err. | Grade −95.00% | Grade +95.00% | *n* |
|---|---|---|---|---|---|---|
| 1 | Included | 6.564103 | 0.204854 | 6.161775 | 6.966430 | 117 |
| 2 | Free | 6.418182 | 0.149391 | 6.124781 | 6.711582 | 220 |
| 3 | Separately | 6.575290 | 0.137685 | 6.304880 | 6.845699 | 259 |

P_paid; LS Means Current effect: $F_{(2, 593)} = 0.33460$, *p* = 0.71576 Effective hypothesis decomposition.

Even though guests that had free parking gave the lowest average grade (M = 6.41) to the importance of stationary traffic as an element of the tourist destination quality, the constructed hypothesis is rejected. Specifically, differences in arithmetic means are not statistically significant. Gained information also leads to a conclusion that users consider the existent prices of parking services as appropriate. This is very important since tourist parking generates a huge revenue in parking fees [54].

The constructed hypothesis suggests that hotel company guests, who evaluate that a hotel company has at its disposal a sufficient number of parking spaces and the possibility to park a passenger car right in front of the hotel company, value the importance of stationary traffic as an element of tourist destination quality. Even when this is so, M = 6.65 (SD = 2.2) vs. M = 6.33 (SD = 2.14) *t*-test results lead to a conclusion that the constructed hypothesis is rejected with a 95% certainty (*t* = 1.75; *p* = 0.8). Therefore, hypothesis H4 is rejected.

Following this, the evaluation of the constructed hypotheses H5, H6, H7, and H8 is given and refers to demographic variables. From the constructed hypotheses only hypothesis H6 and H8 fail to reject. In other words, with regards to demographic variables, sex, and education degree there are no statistically significant differences. In evaluating the importance of stationary traffic as a factor of tourist destination quality, the constructed hypothesis H6 and H8 fail to reject. This result confirms that limited parking skills of women are just a stereotype [55]. The same conclusion applies to both domestic and foreign tourists and therefore hypothesis H5 is to be rejected. It is interesting that both domestic (27.1%) and foreign (25.4%) tourists share the same problems with parking service. A statistically significant difference in evaluating the importance of stationary traffic as a factor of tourist destination quality was determined only against the demographic feature of age (cf. Table 6).

**Table 6.** Difference in evaluating the importance of stationary traffic as a factor of tourist destination quality against the demographic feature of age.

| | Age | Grade-Mean | Grade-Std. Err. | Grade −95.00% | Grade +95.00% | *n* |
|---|---|---|---|---|---|---|
| 1 | 18–25 | 6.014286 | 0.261814 | 5.500087 | 6.528485 | 70 |
| 2 | 26–35 | 6.651515 | 0.190658 | 6.277066 | 7.025965 | 132 |
| 3 | 36–49 | 6.807692 | 0.162370 | 6.488800 | 7.126585 | 182 |
| 4 | 50–64 | 6.534091 | 0.165115 | 6.209808 | 6.858374 | 176 |
| 5 | 65+ | 5.416667 | 0.365082 | 4.699651 | 6.133683 | 36 |

Age; LS Means Current effect: $F_{(4, 591)} = 4.1210$, *p* = 0.00266 Effective hypothesis decomposition.

Based on the information obtained it is evident that survey participants aged from 36 and 49 attach greatest importance to stationary traffic (M = 6.80), and survey participants aged 65+ smallest importance (M = 5.41). This result can be explained by the fact that the older individuals do not travel as much by car on vacation and therefore have less need for parking services [56]. Hypothesis H7 is to be rejected.

Table 7 summarizes the results of the hypotheses testing.

**Table 7.** Hypotheses results.

| Hypotheses | Results |
|---|---|
| H1a: *Tourists that had not experienced a parking problem gave a higher grade to the importance of stationary traffic as a factor of tourist destination quality.* | Failed to reject |
| H1b: *Tourists whose parking problem had been solved in a satisfactory way gave a higher grade to the importance of stationary traffic as a factor of tourist destination quality than tourists whose problem had not been solved in a satisfactory way.* | Rejected |
| H2: *The way of organizing parking impacts the grade of the importance of stationary traffic as a factor of tourist destination quality.* | Rejected |
| H3: *The way of parking payment impacts the grade of the importance of stationary traffic as a factor of tourist destination quality.* | Rejected |
| H4: *The number of parking spaces and a possibility to park the car right in front of the hotel impact the grade of the importance of stationary traffic as a factor of tourist destination quality.* | Rejected |
| H5: *Foreign tourists give a higher average grade to the importance of stationary traffic as a factor of tourist destination quality.* | Rejected |
| H6: *Between men and women there are no statistically significant differences in evaluating the importance of stationary traffic as a factor of tourist destination quality.* | Failed to reject |
| H7: *Between guests of different age there are no statistically significant differences in evaluating the importance of stationary traffic as a factor of tourist destination quality.* | Rejected |
| H8: *Between guests of different education degrees there are no statistically significant differences in evaluating the importance of stationary traffic as a factor of tourist destination quality.* | Failed to reject |

The results show that three out of eight of our hypotheses have failed to reject. Therefore, hypotheses H1, H6, and H8 fail to reject and other hypotheses are rejected.

## 5. Conclusions

More than 19 million tourists visit Croatia every year. Croatia is one of the most desirable European tourist auto-destinations. Except for domestic guests among auto-guests there are tourists from Slovenia, Italy, Austria, and Germany, but also tourists from other European countries. Stationary traffic as an element of tourist destination quality and sustainability is very important particularly to these guests. The conducted research confirmed the importance of stationary traffic as an element of tourist destination quality. Within the grade range from 1 to 10, the upper quartile of survey participants evaluated the importance of stationary traffic as an element of tourist destination quality with grades from 8 to 10. Despite that, results of the conducted research point out the fact that every fourth survey participant already experienced some kind of a parking problem at a tourist destination. Hotels and tourist destination managers should provide a sufficient number of parking spaces for their guests' cars or they risk dissatisfaction with the hotel product and tourist destination as a whole. They can do that in two main different ways. The first one is to dedicate more space to new parking areas and second, to optimize the management of the existing parking areas. The second approach is based on the implementation of smart parking technology and is more environmentally-friendly. Guests at a tourist destination do not expect to have parking problems, which is confirmed by the information that guests, who did not have a parking problem at the tourist destination, attach more importance to stationary traffic as an element of tourist destination quality. However, also the information that guests whose parking problem was solved in a satisfactory way do not give a higher grade to stationary traffic as an element of tourist destination quality than guests whose problem was not solved in a satisfactory way. Building and nurturing a "zero defects" culture is critical for stationary traffic as an element of tourist destination quality and sustainability. The way of organizing parking and of parking payment do not impact the evaluations rate of the importance of stationary traffic as an element of tourist destination quality. As far as demographic variables are concerned, only age impacts the importance

of stationary traffic as an element of tourist destination quality. It is reality to expect that tourist auto-destinations are going to recover much faster from the COVID-19 crisis, which further emphasizes the importance of stationary traffic as an element of quality and sustainability of tourist destinations. In future research, the impact of adequate parking services in tourist destination quality and sustainability in the period during and post COVID-19 pandemic conditions should be investigated.

**Author Contributions:** Conceptualization, D.P., R.M., E.M. and L.K.; methodology, D.P., L.K.; software, D.P.; validation, E.M., and R.M.; formal analysis, D.P.; investigation, R.M., E.M.; resources, R.M.; data curation, R.M., E.M.; writing—original draft preparation, D.P., R.M.; writing—review and editing, E.M., L.K.; visualization, L.K.; supervision, D.P.; project administration, R.M. All authors have read and agreed to the published version of the manuscript.

**Funding:** This research received no external funding.

**Data Availability Statement:** The data presented in this study are available on request from the corresponding author.

**Conflicts of Interest:** The authors declare no conflict of interest.

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
