# Peer review of "Stationary Traffic as a Factor of Tourist Destination Quality and Sustainability"

_sustainability, doi:10.3390/su13073965_

Round 1
Reviewer 1 Report
Dear Authors,
Thank you for inviting me to read this manuscript.
The article is consistent with the lines of the magazine and the topic is certainly of interest to readers. It can be said that the article, on the whole, makes a rather interesting contribution to the debate on the subject in question.
Overall the presentation is reasonably good, but it might still require some work:
- The authors mention the Republic of Croatia as the country where the study took place; I think the authors should introduce the annual number of domestic and foreign tourists earlier.
- I suggest adding a table with the demographic features of domestic guests and foreign guests.
- Figure 2 and Figure 5 is not needed as results are provided in the text already.
- The research methodology, application of appropriate statistical techniques are all adequate for the analysis conducted.
- There is no information in which statistical program the analyzes were performed.
- The paper should have limitations and a future research section at the conclusion.
- I suggest if you can provide the detailed questionnaire used for the survey, data collection in the supplementary it will be great.
Author Response
The article is consistent with the lines of the magazine and the topic is certainly of interest to readers. It can be said that the article, on the whole, makes a rather interesting contribution to the debate on the subject in question.
Overall the presentation is reasonably good, but it might still require some work:
The authors mention the Republic of Croatia as the country where the study took place; I think the authors should introduce the annual number of domestic and foreign tourists earlier.
We did it in the first two sentences of our article and at the end of introduction where we mention the County of Primorje and Gorski kotar and Istria.
I suggest adding a table with the demographic features of domestic guests and foreign guests.
According to yours suggestion we put the table in the part Materials and method.
Figure 2 and Figure 5 is not needed as results are provided in the text already. ď‚· We removed the figure 2 and 5.
The research methodology, application of appropriate statistical techniques are all adequate for the analysis conducted.
There is no information in which statistical program the analyzes were performed.
In the section Materials and method we put the information about statistical program (Statistica) which we used for evaluation of collected data.
The paper should have limitations and a future research section at the conclusion.
We put the recommendation for future research in the conclusion of our paper.
I suggest if you can provide the detailed questionnaire used for the survey, data collection in the supplementary it will be great.
We did it in the section Materials and method.
Reviewer 2 Report
This research could be useful for other car destinatios?
Can you compare with that destinations?
I would like to remember that you have to use the citation style of the journal
Line 68
The task of quality factors of a tourist destination is double 68 (Blazeska, Milenkovski, Gramatnikovski, 2015):
Could you use bullet poihnts? Or a Smart to introduce de most important ideas'
You have said that factors are double, but you show 11 points. Can you add more literature in each one of this 11 points?
Line 131 can you add some literature related with that?
I'll recommend authors separate point 2 in two points. By one side the literature review and focus only in literature that helps to reinforce the idea of the research and the conceptual model. By the other side the concept of the research higlighting the model supported by the literature more clearly.
Line 198 maybe you can add a table for a beter explanation, the same for the main descriptive data of the sample in line 206 and you can add a graph.
Line 235 -236 is literature that support hypothese, if you explain before, now you can focus only in resultsThe same for the rest of the hypothesis
Figure 2 You forget the X-axis and named .
And I can not find the results of the t-test or the p-value related with the analysis. I miss the stata statistics tables
H2 it could be beter explained with the statistical report
I recommend authors add the table results of the statistical analysis
In line 305 you said "
Specifically, differences in arithmetic means 305 are not statistically significant." But we can not see the statistical results
Figure 5 same of figure 2
From line 339 to line 345 you have mixed conclusions and results
Paper needs the statistaical tables of results
Conclusions ara really poor
Round 2
Reviewer 2 Report
Authors have to remember thath If the null hypothesis cannot be disproved it should be stated that we “fail to reject,” rather than “prove” or “accept,” the hypothesis.
Author Response
Dear Sir,
The text has been corrected accordning to your instructions. Thanks for tips.
This manuscript is a resubmission of an earlier submission. The following is a list of the peer review reports and author responses from that submission.
Round 1
Reviewer 1 Report
- Before this study, most people know traffic issues (or parking) are important for tourism and this is almost a common sense. The authors' problems (what's the different with common sense?) and purposes should be express more clearly.
- The authors should dress more about the contributions. And tell us more of this research had discovered.
- From line 169: The study covered guests from 16 hotels and 1 tourist 169 resort... This looks like not random sampling. The authors used sampling in some specific place and infer the statistics results. The scopes of sampling and population do not match. The authors should define the research scope be clearly.
- In Table 1, the numbers of "percent" and "cumulative - percent" are wrong. The comma should be changed to be period.
- In Table 3, "yes" had 154 counts and "no" had 441 counts. I thinks the results should indicate that survey participants that did not agree that not experienced a parking problem gave a higher grade to the importance of stationary traffic as a factor of tourist destination quality. This is different with authors' results from line 214 to 219.
- In H2 and H3, there are some conditions inside hypothesis. Do the authors test the results of hypothesis? Are H2 and H3 accept or reject the hypothesis?
- In H5,H6, and H7, only H6 is accepted. But, from line 283 to 286, it shown sex and education degree there are no statistically significant differences in evaluating the importance of stationary traffic as a factor of tourist destination quality. I think these are wrong. Should be that survey participants that did not agree that "sex and education degree there are no statistically significant differences in evaluating the importance of stationary traffic as a factor of tourist destination quality." These are different meanings.
Reviewer 2 Report
Thank you for the opportunity to review your paper. I am sorry to say that your paper is not yet ready for publication and my comments below are intended to support you in developing your paper further.
- You abstract needs re-writing to make it clearer the purpose of the research and the originality.
- Your introduction and literature review needs re-writing and re-organising the make it clearer what is the contribution to knowledge. You did try to show why parking at the destination is important but this is based on dated literature. What about the literature on destination image, attractiveness and quality? Here you need to show why we need to isolate and study parking in particular as a quality factor. What makes parking an area which needs to be studied? This is missing in your research.
- Please pay attention to your writing style as it was difficult to understand what you were trying to say at some points in the paper.
- Greater justification of your methods and choice of sample is needed. It was only till I reached the methods section did it become clear that you were focusing on hotel parking. Hence the literature review would need to be tailored around this and more on hotel service quality.
- Your discussion and conclusions were limited and they were not interpreted in light of the literature. What practical solutions can you offer to hoteliers ?